# Evolution of antibiotic resistance at low antibiotic concentrations including selection below the minimal selective concentration

Isobel C. Stanton [1], Aimee K. Murray [1], Lihong Zhang [1], Jason Snape[2] & William H. Gaze [1]✉

Determining the selective potential of antibiotics at environmental concentrations is critical for designing effective strategies to limit selection for antibiotic resistance. This study determined the minimal selective concentrations (MSCs) for macrolide and fluoroquinolone antibiotics included on the European Commission's Water Framework Directive's priority hazardous substances Watch List. The macrolides demonstrated positive selection for *ermF* at concentrations 1–2 orders of magnitude greater (>500 and <750 µg/L) than measured environmental concentrations (MECs). Ciprofloxacin illustrated positive selection for *intI1* at concentrations similar to current MECs (>7.8 and <15.6 µg/L). This highlights the need for compound specific assessment of selective potential. In addition, a sub-MSC selective window defined by the minimal increased persistence concentration (MIPC) is described. Differential rates of negative selection (or persistence) were associated with elevated prevalence relative to the no antibiotic control below the MSC. This increased persistence leads to opportunities for further selection over time and risk of human exposure and environmental transmission.

---

[1] European Centre for Environment and Human Health, University of Exeter Medical School, ESI, University of Exeter, Penryn Campus, Penryn, Cornwall TR10 9EF, UK. [2] AstraZeneca, Alderley Edge, Cambridge, Cheshire SK10 4TG, UK. ✉email: W.H.Gaze@exeter.ac.uk

Antibiotic concentrations found in the environment, released from anthropogenic sources[1,2], are lower (ng/L–µg/L)[3] than minimum inhibitory concentrations (MICs). Traditionally these concentrations have not been regarded as posing a risk in terms of selecting for antimicrobial resistance (AMR). However, in research published in 2011 and 2014, single species competition assays determined that selection occurs at concentrations considerably lower than MICs, with the lowest selective concentration (where the resistant strain is enriched over the susceptible) termed the "minimal selective concentration" (MSC)[4,5]. MSCs were determined for various compounds, e.g., 100 ng/L for ciprofloxacin to 3 mg/L for erythromycin, in *Escherichia coli* strains carrying both chromosomal and plasmid borne resistance mechanisms[4,5].

Subsequently, attempts have been made to determine MSCs of antibiotics in complex microbial communities more representative of human, animal and environmental microbiomes. One study investigated the selective potential of tetracycline in a model biofilm[6] establishing that the prevalence of *tetA* and *tetG* tetracycline resistance genes was significantly higher at 1 µg/L compared to a no antibiotic control[6]. The same team also demonstrated significant increase in resistance to ciprofloxacin in *E. coli* isolated from a complex community in a test tube system and in a biofilm system (at 5 and 10 µg/L of ciprofloxacin, respectively) in comparison to a no antibiotic control[7]. A more recent study undertook evolution experiments similar to those used by Gullberg et al.[5] in laboratory batch microcosms, but with a complex microbial community rather than single species inoculum. A MSC of cefotaxime was determined as 0.4 µg/L using qPCR to track prevalence of the $bla_{CTX-M}$ genes over time[8].

Data using both single species and complex community experiments suggest that antibiotic concentrations found in environmental settings may select for AMR[9]. Data has been published showing associations between environmental AMR exposure and negative health outcomes in humans. One study determined a link between surfing, and therefore higher exposure to bathing waters, and increased gut carriage of CTX-M-producing *E. coli* in comparison to non-surfers[10]. It is critical to determine MSCs for antibiotics and co-selective agents, as there is currently no requirement and no agreed test guidelines to test selective potential. Mitigation strategies may be required to reduce selection for AMR in the environment reducing the probability of environmental transmission[11] and evolution of new resistant strains.

In 2015, the European Commission produced a report with a list of 10 priority substances or groups of substances which are potentially detrimental to the aquatic environment and require better monitoring. This list included the three macrolide antibiotics (azithromycin, clarithromycin and erythromycin). To determine which compounds should be placed on the Watch List, predicted no effect concentrations (PNECs) were compared to predicted environmental concentrations (PECs) for all three antibiotics, and measured environmental concentrations (MECs), for azithromycin and clarithromycin. Toxicity data for *Ceriodaphnia dubia*, *Anabaena flos-aquae* and *Synechococcus leopoldenisis* were used to determine PNECs for azithromycin, clarithromycin and erythromycin, respectively. In all cases the PECs and MECs exceeded the PNECs generating unacceptable risk quotients (RQs) > 1[12]. In 2018, the Watch List was updated with the continued inclusion of the macrolides and the addition of antibiotics ciprofloxacin and amoxicillin[13].

Macrolides inhibit protein synthesis in bacterial cells by binding to the 23S rRNA component of the 50S subunit of the ribosome. This prevents newly synthesised peptides passing through the ribosome tunnel and, subsequently, translation[14,15]. There are a range of mechanisms that bacteria employ to resist macrolides including rRNA methylases, efflux pumps (both ATP-binding transporters and major facilitators), esterases and phosphorylases[16]. Macrolides have been detected in a variety of environmental settings. Concentrations range from ng/L to µg/L with a maximum MEC (excluding unusually high concentrations from pharmaceutical production effluents, for example) of 4 µg/L of erythromycin-$H_2O$, a metabolite of erythromycin which is thought to select for resistance genes[17,18], measured in surface water in the Jianhan Plain, China[19].

Fluoroquinolones (FQs) are a synthetic class of broad specturm antibiotics which has led to them being used extensively worldwide[20,21]. In 2012, ciprofloxacin was the most highly prescribed FQ in European countries accounting for 71% of consumption[22,23]. This class of antibiotics works by binding to, and inhibiting, bacterial type II topoisomerases which are important for cellular processes including DNA replication[21]. Due to the extensive FQ use in the clinic, many mechanisms conferring resistance to FQs have emerged[20]. These include mutation of the target site and transferable resistance genes such as *qnr* genes that encode proteins that block the target site[21]. Ciprofloxacin has been measured as high as 31 mg/L in pharmaceutical effluent in India[24], although a median concentration of 0.12 µg/L was calculated using the Umweltbundesamt (German Environment Agency) "Pharmaceuticals in the environment" database (excluding values where ciprofloxacin was below the detection limit)[25]. This is more indicative of typical environmental concentrations of ciprofloxacin.

The aim of this study was to investigate whether current MECs of azithromycin, clarithromycin, erythromycin and ciprofloxacin select for AMR and to determine the MSCs for each compound. Evolution experiments, as previously described[8], were performed with a complex microbial community inoculum in a simple reproducible experimental system with greater bacterial and resistance gene diversity, and therefore realism, than single species model systems. A MSC of tetracycline was also determined to compare this method to the previously published model biofilm system[6]. Providing policy makers and regulators with MSC data is important as this can be used in combination with traditional ecotoxicology data to determine safe discharge levels of antibiotics and other antibacterial compounds, protecting environmental and human health respectively[9,26,27].

Here we show that the three macrolide antibiotics select for AMR at concentrations considerably higher than those found typically in environmental settings but that ciprofloxacin selects for AMR at concentrations more representative of those found in the environment. We also demonstrate a selective window below the MSC which we have termed the minimal increased persistence concentration (MIPC).

## Results

**Assessing the selective potential of macrolides**. Five macrolide resistance targets (*ermB, ermF, mef* family, *mphA* and *msrD*) were selected to quantify with qPCR as they are commonly reported from a range of Gram positive and Gram negative bacteria[16]. In addition, *mphA* was the most common resistance gene found in *E. coli* from clinical samples by Phuc Nguyen et al.[28]. Further, *ermB* and *ermF* were suggested as genetic indicator determinants for assessing resistance to macrolides in the environment[29]. Selection for the *intI1* gene was also determined. *IntI1* encodes the class 1 integron integrase gene. Class 1 integrons have been frequently described as good markers of anthropogenic pollution and of AMR prevalence as they integrate a wide range of antibiotic and biocide resistance genes[30–34].

A review of current macrolide concentrations found in typical aquatic environments (excluding concentrations where unusually

**Table 1 Environmental concentrations of macrolides.**

| Antibiotic | Mean (µg/L) | Maximum (µg/L) |
|---|---|---|
| Azithromycin | 0.193 | 1.5 |
| Clarithromycin | 0.140 | 1 |
| Erythromycin | 0.225 | 2.42 |
| Erythromycin-$H_2O$ | 0.412 | 4 |

high concentrations are found, for example pharmaceutical effluent) for the three compounds, and the metabolite erythromycin-$H_2O$, was undertaken (Table 1) and initial antibiotic concentrations were chosen based on these values. Supplementary Table 1 showing the full list of concentrations and the references can be found in the Supplementary Information.

Initial range finding experiments investigated whether environmentally relevant concentrations (0.1, 1, 10 and 100 µg/L) of macrolides select for the targeted resistance genes, Supplementary Figs. 1–3. No significant positive selection for any of the genes, at any concentration of macrolides, was observed. Investigation of higher concentrations was then undertaken from 1000 to 10,000 µg/L for azithromycin and clarithromycin and at 1000, 10,000 and 100,000 µg/L for erythromycin (as it has been found to be less potent than some of its semi-synthetic derivatives[35]) Supplementary Figs. 4–6.

For *mphA*, significant positive selection at extremely high concentrations was observed (10,000 µg/L for azithromycin and clarithromycin and 100,000 µg/L for erythromycin). Statistically significant positive selection for *ermF* was observed to 90% confidence at 1000 µg/L for azithromycin and erythromycin and to 95% confidence for clarithromycin and at subsequent higher concentrations for all three. No significant positive selection was observed for *ermB*, *msrD* or the *mef* family, although some genes showed increased persistence (i.e., rate of gene loss over the 7 day period was reduced with increasing antibiotic concentration).

This suggested a concentration range between 100 and 1000 µg/L was required to determine more accurate lowest observable effect concentrations (LOECs) and MSCs. A final range of macrolide concentrations was chosen based on responses seen in range finding experiments, these were 100, 250, 500, 750, 1000, 10,000 and 100,000 µg/L. While *ermF*, *mphA* and *intI1* underwent positive selection, only data for *ermF* is presented as a selective effect for this gene was observed at the lowest concentration for all three compounds. However, *mphA* and *intI1* show a much stronger response in terms of greater increases in gene prevalence at higher antibiotic concentrations (Supplementary Figs. 7–12). For all three macrolides, significant positive selection for *ermF* was observed at 750 µg/L (Fig. 1a–c, respectively). Selection for *ermF* was observed at 90% confidence at 750 µg/L by both azithromycin ($p = 0.0616$, $z = −1.541855$, Dunn's test, $\Delta = 5.03$) and erythromycin ($p = 0.0663$, $z = −1.503557$, Dunn's test, $\Delta = 1.89$) and by clarithromycin to 95% confidence ($p = 0.0336$, $z = −1.830510$, Dunn's test, $\Delta = 3.22$) but no significant selection was seen for all of the macrolides at 500 µg/L compared to the no antibiotic control (Fig. 1). We, therefore, determined 750 µg/L as the LOEC for all three macrolides.

For *mphA*, selection by azithromycin occurred at 1000 ($p = 9.21e−5$, $t = 4.470$, Gamma (log) GLM, $\Delta = 67.70$), 10,000 ($p = 0.000413$, $t = 3.941$, Gamma (log) GLM, $\Delta = 43.27$) and 100,000 µg/L ($p = 0.003762$, $t = 3.125$, Gamma (log) GLM, $\Delta = 21.30$). For clarithromycin, no significant increase of *mphA* prevalence, in comparison to the no antibiotic control, was seen until 100,000 µg/L ($p = 0.0446$, $z = −1.699673$, Dunn's test (difference), $\Delta = 8.66$). Similarly, erythromycin did not positively select for *mphA*

until 100,000 µg/L ($p = 0.0361$, $z = −1.797731$, Dunn's test, $\Delta = 26.56$). Graphs for these data can be seen in Supplementary Figs. 7, 9 and 11 for azithromycin, clarithromycin and erythromycin, respectively.

In the presence of azithromycin, *intI1* showed a significant increase, compared to the no antibiotic control, to 90% confidence at 1000 µg/L ($p = 0.0886$, $t = −1.756$, Gamma (inverse) GLM, $\Delta = 415.64$), 10,000 µg/L ($p = 0.0894$, $t = −1.752$ Gamma (inverse) GLM, $\Delta = 288.75$) and 100,000 µg/L ($p = 0.0932$, $t = −1.731$, Gamma (inverse) GLM, $\Delta = 125.42$). An increase in *intI1* prevalence in the presence of clarithromycin, compared to the no antibiotic control, was observed only at 100,000 µg/L ($p = 1.46e−05$, $t = 5.105$, Gaussian GLM (difference), $\Delta = 1.81$). Erythromycin also selected for *intI1*, in comparison to the no antibiotic control, at 100,000 µg/L ($p = 0.0142$, $z = −2.191057$, Dunn's test, $\Delta = 10.63$). Graphs for this data can be seen in Supplementary Figs. 8, 10 and 12 for azithromycin, clarithromycin and erythromycin, respectively.

It was not possible to determine a MSC for selection of *ermF* by azithromycin and clarithromycin as the trendline was always above the *x*-axis for both. The MSC is defined where the line of best fit crosses the *x*-axis (Supplementary Figs. 13 and 14, respectively). A MSC of erythromycin, however, was calculated (Fig. 2) and was 514.1 µg/L for *ermF*.

To determine if mutation based resistance to macrolides occurred below the LOEC, phenotypic resistance was quantified at a range of azithromycin concentrations. No significant selection for resistance was observed for *Enterobacteriaceae* spp. on Chromocult agar, *Staphylococci* spp. on Mannitol-salt agar or bacteria able to grow on Mueller-Hinton agar at 100 µg/L. Although some increase in resistance was observed at 1000 µg/L, this was not significant (Supplementary Fig. 15).

Metagenome analysis was undertaken on a subset of samples from macrolide selection experiments as this was the main focus of the study and has not been investigated by previous studies. Metagenome analysis of tetracycline and ciprofloxacin selection has, however, been previously investigated in studies by Lundström et al.[6] and Kraupner et al.[7]. Three replicates were taken from each treatment including the LOEC for all three macrolides (750 µg/L), a concentration below this (250 µg/L) and concentrations higher than this where a strong selective effect is seen by *intI1* (1000, 10,000 and 100,000 µg/L).

Metagenome analysis enabled the relative abundance of all characterised macrolide–lincosamide–streptogramin (MLS) resistance genes to be determined as a function of macrolide concentration (Fig. 3). A significant difference in MLS gene prevalence was observed for both azithromycin ($p = 0.0280$, $z = −1.911797$, Dunn's test, $\Delta = 8.84$) and erythromycin ($p = 0.0843$, $z = −1.376494$, Dunn's test, $\Delta = 0.72$) at 10,000 but not for clarithromycin. A significant difference was seen at 100,000 µg/L and for azithromycin ($p = 0.0047$, $z = −2.600044$, Dunn's test, $\Delta = 15.72$) clarithromycin ($p = 0.0089$, $z = −2.370629$, Dunn's test, $\Delta = 5.16$) and erythromycin ($p = 0.0109$, $z = −2.294157$, Dunn's test, $\Delta = 5.40$) compared to the no antibiotic control.

We also observed some individual macrolide genes increasing in prevalence, compared to the no antibiotic control, at 250 µg/L, Supplementary Figs. 16–18. This is currently lower than our LOEC and MSC defined by *ermF*. We, therefore, quantified molecular prevalence of two of these genes (*ermB* and *macB*) with qPCR, as it has been deemed to be a more sensitive approach than metagenomics and considers gene prevalence in the entire community rather than just the sequenced fraction[6]. These genes were chosen to represent the resistance genes observed increasing in relative abundance at lower concentrations. *MacA* was not quantified as it is always found in conjunction with *macB* as they

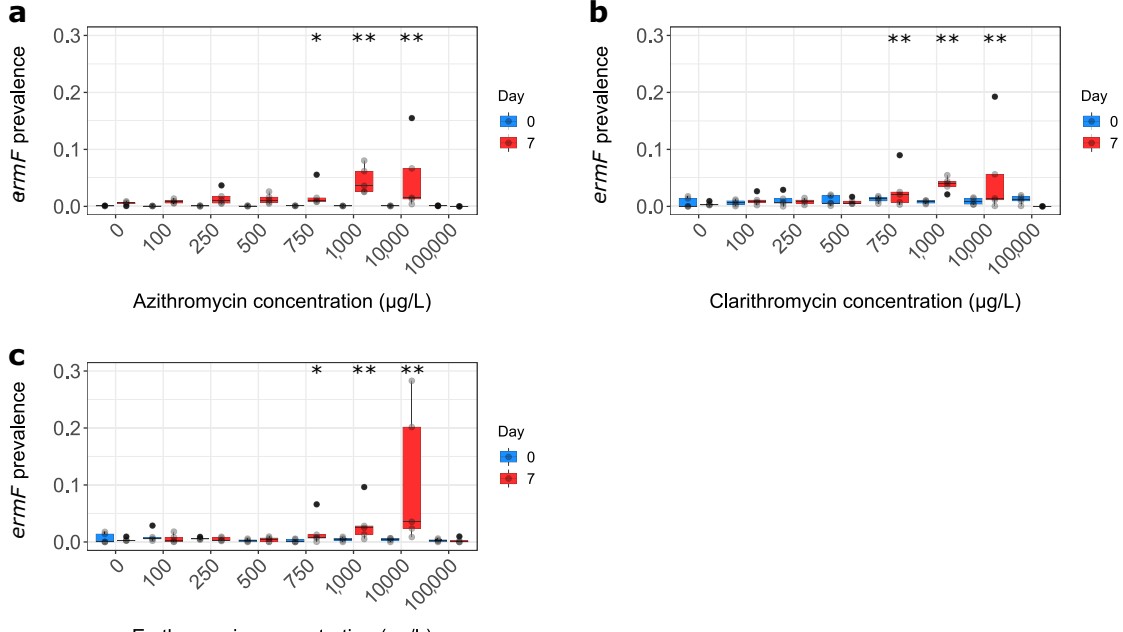

**Fig. 1 Selection for *ermF* by macrolide antibiotics. a** Azithromycin. **b** Clarithromycin. **c** Erythromycin. *Significant positive selection to 90% confidence in comparison to the no antibiotic control. **Significant positive selection to 95% confidence in comparison to the no antibiotic control. $n = 5$ replicates per concentration. One high outlier replicate has been removed from the clarithromycin experiment (day 7, 250 μg/L). Boxplots follow the Tukey's representation.

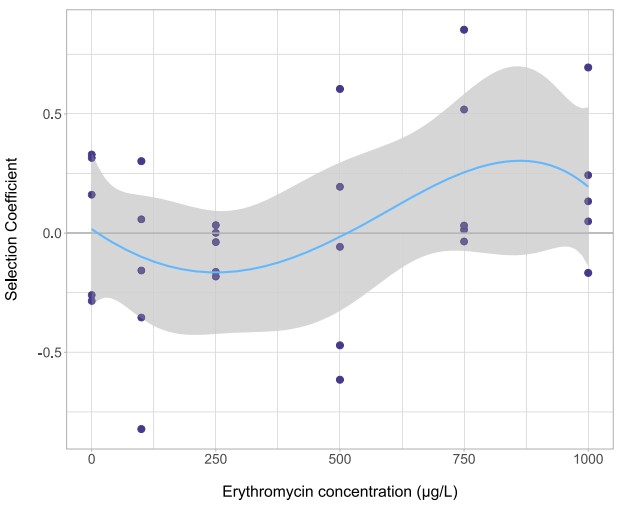

**Fig. 2 Selection coefficient graph for *ermF* by erythromycin.** Selection coefficient values were determined as described previously[5]. These were plotted with a line of best fit (polynomial regression line order 4, $R^2 = 0.1709$, $y = -2.544e^{-12}x^4 + 1.564e^{-09}x^3 + 2.59e^{-06}x^2 - 0.001432x + 0.01684$). Here the line of best fit crosses the x-axis at 514.1 μg/L and this is defined as the MSC for this gene selected for by this compound. $n = 5$ replicates per concentration.

encode 2 subunits of an ABC-type efflux pump—MacAB[36,37]. Using qPCR, no positive selection was observed for these genes (i.e., the prevalence of these genes did not increase over time in comparison to the no antibiotic control), Supplementary Figs. 19 and 20.

Co-selection was observed at high concentrations of azithromycin and clarithromycin, Supplementary Figs. 21–23. Resistance to certain antibiotic classes appeared to be selected for at relatively low concentrations of erythromycin (250 μg/L) although if individual gene abundances were compared, a dose dependent response of prevalence did not demonstrate an association with antibiotic concentration until much higher concentrations suggesting this may be an artefact (Supplementary Fig. 24).

The metagenome analyses also provided information on community structure. Replicates from all treatments were found to be dominated by *E. coli* and unclassified *Escherichia* spp. but also included a range of other Gram negative and Gram positive taxa.

Replicates treated with azithromycin and clarithromycin became less diverse with increasing concentration of antibiotic, but there was a less clear pattern when samples were treated with erythromycin. Many species were undetectable when samples were treated with 100,000 μg/L of the specific macrolide, indicating exposure to high concentrations reduced community diversity (Supplementary Figs. 25–27).

**Assessing the selective potential of ciprofloxacin**. The *Enterobacteriaceae* clinical breakpoint from the EUCAST database was chosen as the maximum concentration for ciprofloxacin—1000 μg/L (EUCAST, Clinical breakpoints—bacteria (v. 4)). Subsequent concentrations were a twofold dilution series down to 0.98 μg/L.

*qnrS* was the class specific gene targeted by qPCR to investigate selection by ciprofloxacin at a range of concentrations. It is the most common gene identified from clinical *Enterobacteriaceae* isolates, is mobile and is often found in environmental strains[21,38]. In addition, *qnr* genes have been reported embedded in complex class 1 integrons[39]. The *intI1* gene was also enumerated to investigate selection by ciprofloxacin. No significant selection for *qnrS* was observed at any concentration (Supplementary Fig. 28). For the *intI1* dataset, the Dunnett's/Dunn's approach did not align well to the biological effect. No significant selection was observed with the Dunn's test until 125 μg/L, however a clear biological effect can be seen at 15.625 μg/L. For this reason, GLM was used. This determined a significant

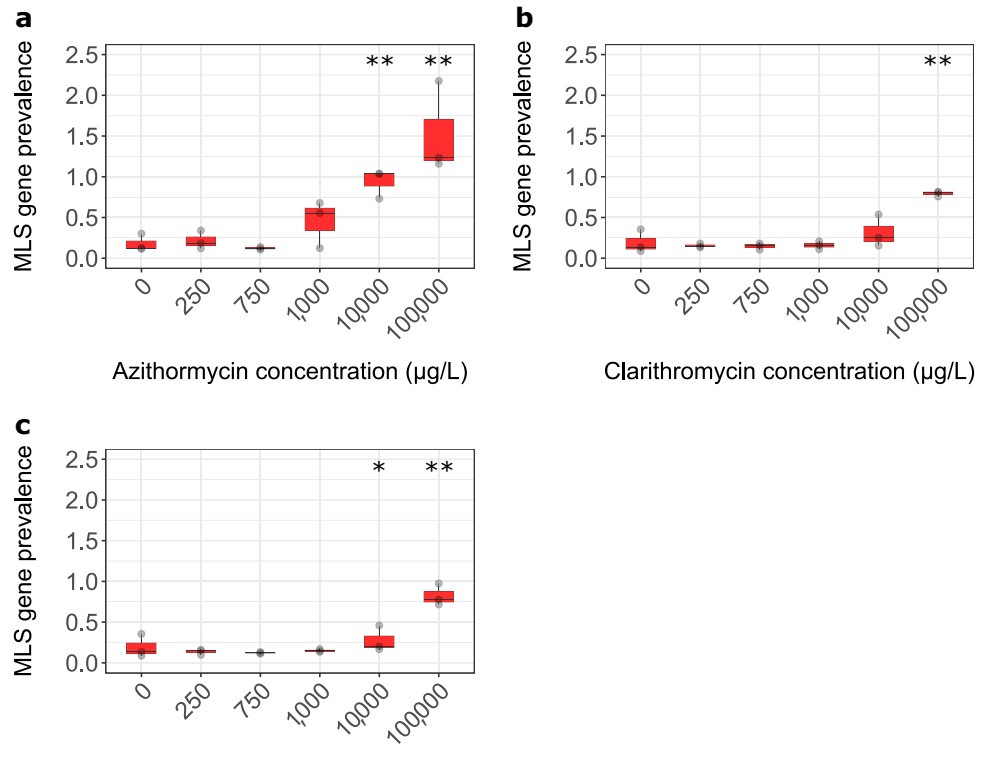

**Fig. 3 MLS resistance gene prevalence as a function of macrolide concentration. a** Azithromycin. **b** Clarithromycin. **c** Erythromycin. *Significant increase to 90% confidence in comparison to the no antibiotic control. **Significant increase to 95% confidence in comparison to the no antibiotic control. $n = 3$ replicates per concentration. Boxplots follow the Tukey's representation.

effect to 90% confidence at 15.625 ($p = 0.0634$, $t = 1.901$, GLM Gamma (identity), $\Delta = 14.81$) and 31.25 µg/L ($p = 0.0553$, $t = 1.964$, GLM Gamma (identity), $\Delta = 21.01$) and to 95% confidence at 62.5 ($p = 0.0491$, $t = 2.019$, GLM Gamma (identity), $\Delta = 32.39$), 125 ($p = 0.0470$, $t = 2.039$, GLM Gamma (identity), $\Delta = 40.04$), 250 ($p = 0.0437$, $t = 2.071$, GLM Gamma (identity), $\Delta = 63.89$), 500 ($p = 0.0438$, $t = 2.070$, GLM Gamma (identity), $\Delta = 62.75$) and 1000 µg/L ($p = 0.0429$, $t = 2.080$, GLM Gamma (identity), $\Delta = 75.78$) (Fig. 4).

By plotting the data as selection coefficients, a MSC of 10.77 µg/L was determined (Fig. 5).

**Comparing in vitro assays for determining MSCs.** To determine whether MSCs/LOECs calculated here were comparable to the biofilm microcosm assay developed by Lundström et al.[6], *tetG* was quantified using qPCR in a selection experiment run for 7 days under tetracycline hydrochloride selection. Tetracycline concentrations were selected to span concentrations where the MSC was reported by Lundström et al.[6] (i.e., 0.1, 1, 10 and 100 µg/L). Significant selection to 90% confidence was observed at 1 µg/L ($p = 0.0784$, $z = -1.416214$ Dunn's test, $\Delta = 2.01$), 10 µg/L ($p = 0.0658$, $z = -1.507583$, Dunn's test, $\Delta = 13.84$) and 100 µg/L ($p = 0.0784$, $z = -1.416214$, Dunn's test, $\Delta = 11.42$) in comparison to the no antibiotic control for *tetG* at day 7 (Fig. 6a). However, when data was analysed by comparing prevalence at day 0–7 for each concentration as previously in this study, loss of the *tetG* genes at all tetracycline concentrations tested (0.1–100 µg/L) was observed (Fig. 6b). The average starting prevalence of *tetG* in the current study was 0.0096 and the highest prevalence at the end of the 7 days was $4.3E^{-06}$. In the selection coefficient graph produced for this dataset, no positive selection was

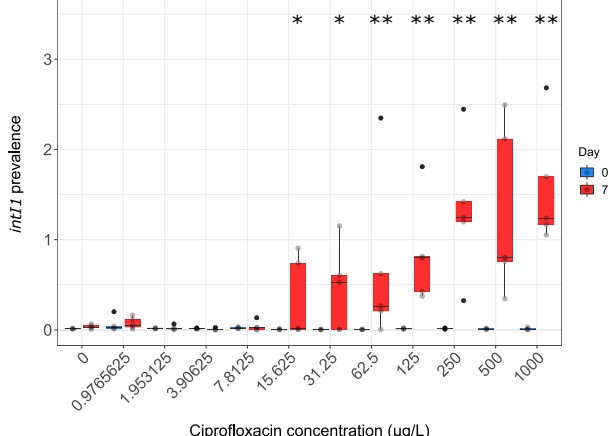

**Fig. 4 Selection for *intI1* by Ciprofloxacin.** Significant selection for *intI1* by ciprofloxacin is seen at concentrations of 15.625 µg/L ($p = 0.0634$, Gamma GLM) and higher. *Significant positive selection to 90% confidence in comparison to the no antibiotic control. **Significant positive selection to 95% confidence in comparison to the no antibiotic control. $n = 5$ replicates per concentration. Boxplot follows the Tukey's representation.

observed so a MSC could not be determined (Supplementary Fig. 29).

**Discussion**

The data generated in this study suggests that, in this experimental system, environmental concentrations of the three macrolides do not positively select for macrolide resistance genes.

For all of the macrolides, significant positive selection for *ermF* was observed at 750 µg/L but not at 500 µg/L.

The LOECs determined here are significantly higher than the maximum MECs determined by the literature review (Table 1) (1.5, 1, 2.4 and 4 µg/L for azithromycin, clarithromycin, erythromycin and erythromycin-H$_2$O, respectively).

By applying a tenfold assessment factor (as recommended by the European Medicines Agency[40]) to 500 µg/L (i.e., the highest no observable effect concentration (NOEC)) for azithromycin, clarithromycin and erythromycin, a PNEC of 50 µg/L was obtained. For erythromycin, a tenfold assessment factor can be applied to the MSC (514.1 µg/L) to calculate a PNEC of 51.41 µg/L. However, as macrolide resistance mechanisms developed by bacteria are common to all three macrolides, we assume that they will have an additive selective effect when all three compounds are released together (although this has not been tested and should be considered in future studies). These PNECs may still, therefore, be underestimates when taking into account combined exposure effects.

In the case of selection for *intI1* and *mphA* using qPCR and the MLS genes from the metagenome analysis, azithromycin appears to be more selective than both clarithromycin and erythromycin, whereas the latter two appear to correlate with each other. One possible explanation is that clarithromycin and erythromycin are more chemically similar to each other, containing a 14 member lactone ring, whereas azithromycin contains a 15 member lactone ring[41]. In addition, it has been shown that azithromycin is a more potent drug than erythromycin[35]. Furthermore, one study demonstrated lower MICs for azithromycin in comparison to erythromycin and found it had increased potency in a range of different bacterial species[42].

MSCs and PNECs generated in this study are significantly higher than the estimated PNECs for the selection of resistance (PNEC$^R$s) calculated by Bengtsson-Palme and Larsson[43] (azithromycin (0.25 µg/L), clarithromycin (0.25 µg/L) and erythromycin (1 µg/L))[43], the freshwater PNECs reported by the European Commission[12] (azithromycin (0.09 µg/L), clarithromycin (0.13 µg/L) and erythromycin (0.2 µg/L))[12] and the PNEC in surface water determined by Le Page et al. (azithromycin (0.019 µg/L), clarithromycin (0.084 µg/L) and erythromycin (0.2 µg/L))[26]; but lie in between the MSCs determined for erythromycin for chromosomal and plasmid based resistance (200 µg/L and 3000 µg/L, respectively) determined in single species assays by Gullberg et al.[4]. This suggests that current ecological PNECs may be protective of resistance selection for macrolides, but this may not be the case for all classes of antibiotics[8,26]. The reasons behind these variations in selective effect concentrations are complex. Gullberg et al.[4] demonstrated that resistance mechanism (e.g., location of mutation) and genetic context influenced MSCs in a single host species system. It is also likely that host identity affects MSC and Klümper et al.[44] recently demonstrated that when a focal *E. coli* strain was embedded within a complex microbial community the MSC increased by 13–43 times[44]. Therefore, higher observed MSCs in complex microbial communities, compared to single species assays, are likely to be driven by biological processes as well as by less sensitive detection methodologies used (e.g., flow cytometry of fluorescently labelled isogenic strains compared to qPCR and metagenomic approaches) and greater variation between replicates due to the complexity of the system.

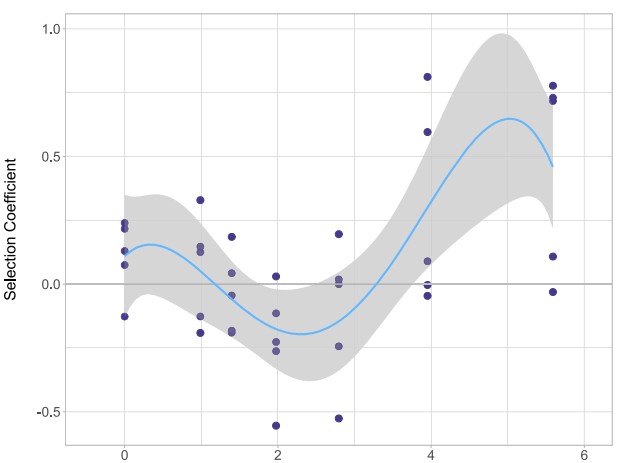

**Fig. 5 Selection coefficient graph for *intI1* by ciprofloxacin.** Selection coefficient values were determined as previously in Gullberg et al.[5]. These were plotted with a line of best fit (polynomial regression line, order 4, $R^2 = 0.4396$, $y = 0.1093 + 0.293x - 0.5274x^2 + 0.1921x^3 - 0.0188x^4$). Here the line of best fit crosses the *x*-axis at 10.77 µg/L and this is defined as the MSC. Plotted here is a square root transformation of the ciprofloxacin concentrations 0.9765625, 1.953125, 3.90625, 7.8125, 15.625 and 31.25 µg/L. *n* = 5 replicates per concentration.

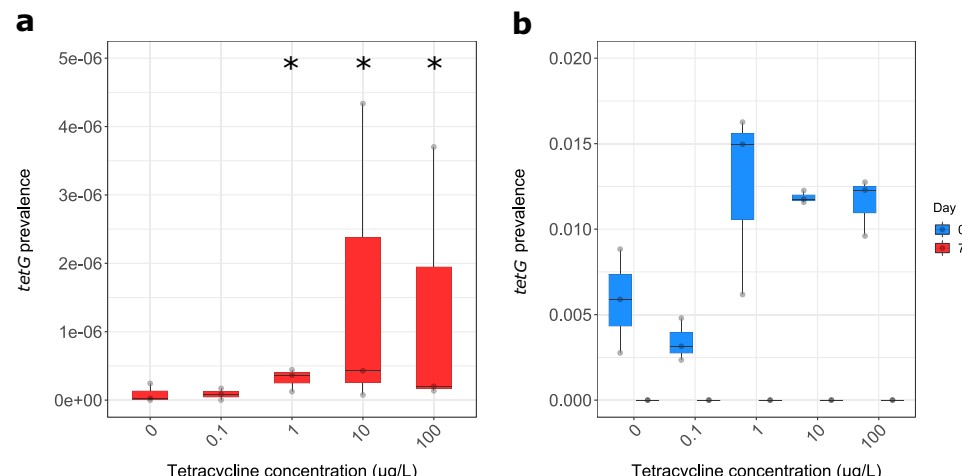

**Fig. 6 Persistence of *tetG* as a function of tetracycline concentration. a** Persistence of *tetG* at day 7. **b** *tetG* prevalence at both day 0 and day 7. *Significant increase to 90% confidence in comparison to the no antibiotic control. *n* = 3 replicates per concentration. Boxplots follow the Tukey's representation.

The decrease in diversity of species observed in the metagenome for all three macrolides, especially at 100,000 μg/L, could explain why a significant decrease in prevalence was seen for *ermF* at 100,000 μg/L. Presumably, the bacterial species predominantly harbouring this gene were significantly reduced in number by this concentration of macrolide.

Whilst the experimental population is dominated by *E. coli* and *Escherichia* spp, there is still a diverse population of bacterial species present. This is not unexpected as the inoculum used was raw wastewater and *E. coli* is a faecal coliform bacterium[45]. The laboratory conditions that these experiments were undertaken in also favour the growth conditions of *E. coli* and other *Escherichia* spp. These species are Gram negative opportunistic pathogens and are, therefore, of great concern in regards to the emergence of resistance. In addition, *E. coli* has been shown to be a reservoir for the macrolide resistance gene *mphA*[28], which was consistently the resistance gene found to be one of the most abundant genes in all three metagenome datasets.

Of all the class specific genes tested, three targets (*ermB, msrD* and the *mef* gene family) did not undergo positive selection at any antibiotic concentration, even at clinically relevant concentrations. One possible explanation might be the low prevalence of these genes in the population in our initial inoculum, although *ermF* consistently demonstrated the lowest starting prevalence of all macrolide resistance genes quantified (except for msrD, which was 0.002 lower) but still demonstrated enrichment with increasing macrolide concentration. It is also possible that the bacterial taxa carrying *ermB, msrD* and *mef* were outcompeted by other resistant taxa with intrinsic or acquired resistance conferred by other mechanisms.

For ciprofloxacin, we determined a MSC of 10.77 μg/L, and with an assessment factor of 10 applied, a PNEC of 1.077 μg/L. These values are in the same order of magnitude as ciprofloxacin MECs reported in aquatic environments (not including pharmaceutical manufacturing waste pollution)[46]. Ciprofloxacin levels in hospital wastewater influent in Switzerland, for example, have been reported between 3 and 87 μg/L[47]. This means that selection for FQ resistance may occur in certain environmental settings polluted with particularly high levels of ciprofloxacin. Furthermore, as this MSC is based on *intI1* selection, it is likely genes conferring resistance to different antimicrobials may also be co-selected by ciprofloxacin at these low concentrations. This is due to the fact that first; some class 1 integron backbones also contain the *sul1* gene (which confers resistance to sulphonamides) and the partly functional, multi-drug efflux pump *qacΔ1* (which confers resistance to quaternary ammonium compounds)*;* and second; class 1 integron arrays are known to carry a variety of different AMR gene cassettes[48]. A class specific qPCR gene target (*qnrS*) was also enumerated, but no selection was observed. Previously, metagenome analyses have been performed in the same experimental system, where untreated wastewater was exposed to ciprofloxacin at 500 μg/L. Even at this comparatively high concentration, no significant increase in known FQ resistance genes was observed. However, prevalence of genes conferring resistance to several other antibiotic classes did increase significantly[49]. This suggests that there may be uncharacterised FQ resistance mechanisms that are selected for below the MSC established by *intI1* prevalence in this study.

The ciprofloxacin MSC determined here (10.77 μg/L) is similar to the LOEC determined in the biofilm (10 μg/L) and test tube (5 μg/L) experiments by Kraupner et al.[7] but are, as with the macrolide compounds, higher than the PNEC[R] calculated by Bengtsson-Palme and Larsson[43] (0.064 μg/L) and the freshwater PNEC calculated by the European Commission[12] (0.089 μg/L). The value determined by the European Commission[12] has, however, had an assessment factor of 50 applied meaning the

NOEC was 4.45 μg/L, which is in the same order of magnitude as the MSC determined in the current study. It is also in the same order of magnitude as the MSCs determined for a variety of chromosomal mutations, by Gullberg et al.[5], that ranged from 0.1 to 2.5 μg/L and the PNEC in surface water determined for ciprofloxacin (0.565 μg/L) by Le Page et al.[26] for cyanobacteria. This agreement in selective effect concentrations across several studies by different research groups suggests that, for ciprofloxacin at least, we can be fairly confident that positive selection occurs in the range of current MECs. The system used here maximises numbers of bacterial generations, and therefore opportunities to observe selection, using high temperature and nutrient conditions. It still, however, generated comparable data to the lower temperature/nutrient flow through biofilm system used previously[6].

A comparison of the method used in this study and that used previously by Lundström et al.[6] was made by undertaking a tetracycline selection experiment. A significant increase in prevalence of *tetG* was observed at 1 μg/L compared to the no antibiotic control (Fig. 6a) when considering only day 7 prevalence (at the end of the experiment), as reported in Lundström et al.[6]. However, when taking into account the starting prevalence, a reduction in *tetG* prevalence over time was observed at all concentrations of tetracycline tested (Fig. 6b). Therefore what was described by Lundström et al.[6] may have been due to increased persistence (i.e., reduced rate of negative selection) and not positive selection or enrichment as suggested. The term MSC should be reserved for the lowest concentration of antibiotic "where the resistant mutant is enriched over the susceptible strain"[50]. A concentration above which a significant increase in persistence is observed could instead be defined as the minimal increased persistence concentration (MIPC). The MIPC is important as concentrations of antibiotic above this will decrease the rate at which resistant bacteria disappear from the environment. This will result in an increased human exposure risk and the probability of subsequent evolution in comparison to environments where no antibiotics are present. It is less of a concern, however, than if positive selection was occurring where numbers of resistance genes and resistant bacteria increase over time. This raises concerns regarding a sub-MSC persistence window (Fig. 7) where numbers of resistant bacteria are higher than if there was no antibiotic present, even though concentrations are below the MSC. Therefore, it could be argued that regulators should be using the MIPC rather than MSC/LOEC as the endpoint when determining safe discharge limits for antimicrobials. The vastly different MSCs/LOECs for the different antibiotics determined here demonstrates the importance of individually testing the selective potential of all antibiotics and other co-selective compounds. It is important that gene targets used to determine selection endpoints by qPCR are appropriate and this can be ensured using metagenome analysis to identify genes enriched at the lowest antibiotic concentrations[8]. However, the existence of uncharacterised resistance genes cannot be ruled out, which is why phenotypic characterisation is still useful.

Although determining the MSC is important for evaluating the selective effects of existing and new antibiotics, it should be used in combination with other bacterial and ecological endpoints. This will enable a more informed assessment of the risk these compounds pose to the environment and indirectly to human health through selection for AMR, as the MSC may not always be the most protective endpoint. In addition it should be noted that the MSC determines the threshold at which positive selection occurs and does not give any insights into the magnitude of the selective effect. For example the macrolide MSC/LOECs were determined by *ermF* but the increase in prevalence was small, whereas the LOEC determined for *intI1* and *mphA* was higher but was associated with a much greater increase in prevalence.

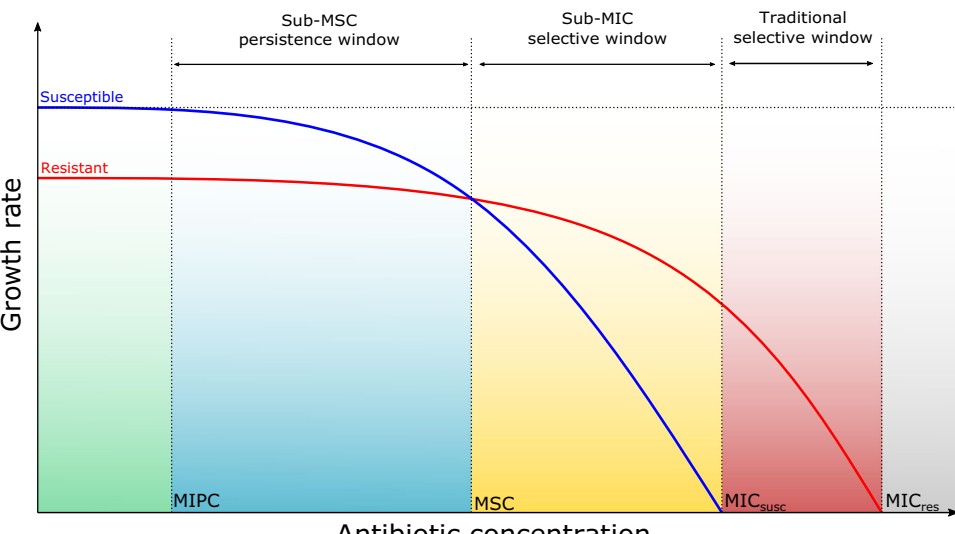

**Fig. 7 Effect of antibiotic concentration on the growth rate of bacteria.** Diagrammatic representation of change of growth rate of susceptible bacteria (blue line) and resistant bacteria (red line) with increasing antibiotic concentration. Graph adapted from Gullberg et al.[5] to include a sub-MSC persistence window (blue area), the area between the MIPC and the MSC. In both the green and blue area, the susceptible bacteria outcompete resistant bacteria. In the sub-MSC persistence window (blue area), however, as growth rate of susceptible begins to decrease there is an overall change in the ratio of resistant to susceptible bacteria and subsequently a difference in total number of resistant bacteria compared to if no antibiotic was present. It is not until the MSC and then in the sub-MIC selective window (yellow area) that resistant bacteria are enriched over susceptible bacteria and positive selection for resistance occurs.

**Table 2 Primer sequences for qPCR.**

| Gene | Forward (5′–3′) | Reverse (5′–3′) | Reference |
|---|---|---|---|
| *ermB* | TTGGATATTCACCGAACACTAGGG | ATAGACAATACTTGCTCATAAGTAACGG | 62 |
| *ermF* | TCTGGGAGGTTCCATTGTCCT | ACTTTCAGGACCTACCTCATAGA | This study |
| *mef* family | GGTGTRYTAGTGGATCGTCA | GMHCCAGCTGCTGCKATAAT | This study |
| *mphA* | TGGTGCATGGCGATCTCTAC | GACGCGCTCCGTGTTGTC | This study |
| *msrD* | CAAGCTGCARAATACGAACAATTT | CCGCAGCCCYTCCAAT | This study |
| *tetG* | GCTAACGAGCCTCACCAAT | TGCGAATGGTCTGCGTAGTA | 6 |
| *intI1* | GCCTTGATGTTACCCGAGAG | GATCGGTCGAATGCGTGT | 63 |
| *16S rRNA* | CGGTGAATACGTTCYCGG | GGWTACCTTGTTACGACT | 64 |
| *qnrS* | CGACGTGCTAACTTGCGTGA | GGCATTGTTGGAAACTTGCA | 65 |
| *macB subtype 1* | CTGCCGTCTCGCAAAACCT | GCACTGGCAGCAACATCAAC | This study |
| *macB subtype 2* | CCCACACTCGAAGCGCTTTA | TGTTGTGGGCGTGGTGGAAG | This study |
| *macB subtype 3* | GCGTCAGCCACCTGTACTTCA | TGAACCAGCTGTACTACGTCG | This study |
| *macB subtype 4* | GCAGCAACACCATCGACATCTA | ACAACACCAGGGTTTCAATCG | This study |

We also show, for the first time to our knowledge, that sub-MSC concentrations, above the MIPC may also be important to consider as they are likely to be associated with increased human exposure risk, increased probability of resistance gene transfer and increased AMR evolution due to greater numbers of resistant bacteria being present in environmental settings. However, with increased persistence the number of resistant bacteria or resistance genes will decrease over time at antibiotic concentrations above the MIPC whereas enrichment through positive selection (at antibiotic concentrations above the MSC) will lead to increased AMR over time, so the two phenomena are fundamentally different in terms of outcome. If the MIPC was used as the selective endpoint when determining safe release limits, this could decrease the PNECs of some antibiotics considerably.

Based on selection for AMR alone, this data would suggest that the macrolides could be removed from the European Commission's Water Framework Directive's priority hazardous substances Watch List, whereas ciprofloxacin should remain.

However, the decision as to whether the macrolides remain on the Watch List, are deprioritised, or are included as priority substances will need to be based on whether the Watch List monitoring data indicates that they pose an EU-wide ecological risk or not.

## Methods

**Complex community sample collection.** A grab sample of raw wastewater influent was obtained from a small wastewater treatment plant in Falmouth, UK serving a population of circa 43,000 in October 2015. Samples were frozen at −80 °C in 40 mL aliquots consisting of 20 mL wastewater and 20 mL 40% glycerol (Fisher).

**Selection experiment.** Samples were washed by pelleting bacteria by centrifugation at 3500 rpm, removing the supernatant and resuspending in the same volume of 0.85% saline solution. This was repeated to remove existing, potentially selective compounds. Iso-sensitest broth (Oxoid) was inoculated with 10% v/v of the washed wastewater sample with the appropriate concentration of antibiotic (azithromycin (Sigma-aldrich), clarithromycin (Molekula), erythromycin (Acros Organics), ciprofloxacin (Sigma-Aldrich) or tetracycline hydrochloride (Fisher)). These were incubated at 37 °C for 24 h, after which 50 μL of the culture was passaged in fresh

broth (5 mL) with the appropriate antibiotic concentrations. These experiments were carried out over a 7 day period with passage every 24 h. In total, 1 mL of samples was taken at day 0 and 7, centrifuged at 14,800 rpm for 3 min and the pellet was resuspended in 1 mL of 20% glycerol and frozen at −80 °C.

**DNA extraction**. DNA was extracted from samples using the MO Bio UltraClean® Microbial DNA Isolation Kit (now QIAGEN DNeasy UltraClean Microbial Kit, 12224-250), as per the manufacturer's instructions. DNA was stored at −20 °C until use.

**Real-time PCR**. The class specific macrolide resistance gene targets chosen were *ermB*, *ermF*, *mphA*, *msrD* and *mef* family (which targeted genes *mefA*, *mefE*, *mefI* and *mefO*), for ciprofloxacin the target was *qnrS* and for tetracycline the class specific gene was *tetG*. The *intI1* integrase gene was also targeted for both the macrolide and ciprofloxacin experiment. Finally, the 16S rRNA gene, which has been used as a proxy for bacterial cell number[6,8], was also enumerated to determine molecular prevalence (target gene copy number/16S rRNA copy number).

Genes tested and their corresponding primers are shown in Table 2.

qPCR using DNA extracted from the macrolide and tetracycline experiments was undertaken using Brilliant III Ultra-Fast Sybr® Green QPCR Master Mix (Agilent Technologies) on the Applied Biosystems StepOne™ machine. Cycling conditions used included an initial cycling stage of 95 °C for 20 s, followed by 50 cycles of 95 °C for 10 s and 60 °C for 30 s. Reactions consisted of 10 µL of master mix, 2 µL of primer pair (10 µM for all primers except 16S which was 9 µM), 0.6 µL of ROX dye, 5 µL of diluted template and were made up to 20 µL with sterile water.

DNA extracted from the ciprofloxacin evolution experiment was analysed using the PrimerDesign PrecisionPLUS MasterMix with pre-added ROX (PrimerDesign). In total, 20 µL reactions consisted of 5 µL of diluted template, 10 µL of mastermix, 2 µL of primer pair (4.5 µM), 0.2 µL of BSA (20 mg/mL) and filter sterilised water up to 20 µL total volume. Cycling conditions used were 10 min at 95 °C, 40 cycles of 95 °C for 15 s and 60 °C for 1 min.

The two mastermixes were compared and no significant difference was seen between copy numbers determined for the sample template DNA.

**Plating experiment**. As azithromycin has been shown to be more potent than the natural compound, erythromycin[35] and has a lower MIC[42], a phenotypic resistance experiment was only conducted on azithromycin as it was expected to show the lowest response of the three macrolides.

Day 7 cultures from samples grown with azithromycin concentrations of 0, 100, 1000 and 10,000 µg/L were plated onto three different types of agar to determine phenotypic resistance. These were Chromocult Coliform Agar Enhanced Selectivity (Merck), for growing *Enterobacteriaceae* spp., Mannitol-salt agar (composition according to HiMedia Laboratories Technical Data protocol), for *Staphylococcus* spp., and the non-selective (in terms of bacterial diversity) Mueller-Hinton agar (Oxoid). Serial dilutions of 100 µL of culture were plated onto agar with and without azithromycin. For Chromocult agar, 16 mg/L azithromycin was used (the clinical breakpoint for *Salmonella* Typhi and *Shigella* spp. (EUCAST, Clinical breakpoints—bacteria (v 7.1)), and for Mannitol-salt and Muller-Hinton agar 2 mg/L was used (the clinical breakpoint for *Staphylococcus* spp. (EUCAST, Clinical breakpoints—bacteria (v. 7.1))).

**Metagenome sequencing**. A subset of samples from the week long selection experiments were chosen for metagenomic analysis. Three replicates from 0, 250, 750, 1000, 10,000 and 100,000 µg/L for all 3 macrolides were sequenced.

DNA was purified as described in Murray et al.[8] using RNase A (Qiagen) and AMPure XP beads (Beckman Coulter). Samples were sent to Exeter Sequencing Centre. Libraries were prepared using the Nextera XT DNA Library Prep Kit and paired end sequencing was undertaken on a HiSeq 2500.

Sequences were first trimmed for adaptor removal using Skewer[51] and quality was checked using FastQC[52] and MultiQC[53]. Paired end reads were combined using FLASH version 2[54] and MetaPlAn2[55] was used to assign bacterial species. Heatmaps for species diversity were generated using Hclust2[56]. Antibiotic Resistance Gene Online Analysis Pipeline (ARGs-OAP) version 2[57] was used, with default settings, to quantify relative abundance and diversity of AMR genes.

**Statistics and reproducibility**. The macrolide range finding experiments and the tetracycline experiments have three biological replicates per concentration. All other experiments had five biological replicates per concentration.

All statistical analyses were performed in R Studio[58]. ANOVA and Dunnett's tests[59] were performed for parametric data; and for non-parametric data, Kruskal Wallis and Dunn's tests[60] were undertaken. Significance was determined to 90 and 95% confidence. As mixed community experiments are inherently noisy due to founder effects and stochasticity, 90% confidence was also highlighted to show less strong associations and will be more protective of selection for AMR in the environment. Where the Dunnett's/Dunn's test did not align well to the biological effect observed, a general linearised model (GLM) approach was used. Gaussian and Gamma model families were explored with various link functions and the best model fit was selected if assumptions were well met (if residuals were normal and variances were homogenous) and by testing for overdispersion. Test statistics are

reported (z for Dunn's test and t for GMLs) and Glass's delta (Δ) is used to report estimated effect size.

ANOVA/Kruskal Wallis tests were performed on the prevalence at day 0 to ensure there was no variation between starting prevalences of samples. If a difference was observed, the post-hoc tests described above were undertaken on the difference between day 0 and 7. This is specified where appropriate.

Selection coefficient graphs were produced using the formula $s = [\ln$ (prevalence at day 7/prevalence at day 0)]/7 as in Gullberg et al.[5]. Where genes were determined by qPCR to have a copy number of zero at day 0, we determined a pseudo value as these samples always showed presence of the gene at day 7. We were able to conclude, therefore, that these genes were below the limit of detection at the start of the experiment. The pseudo value was taken to be half of the detection limit as an estimate. A line of best fit (testing linear and polynomial models) was determined using the polynom package[61] in R studio and a summary of the models was produced. The model with the best fit was determined by considering the $R^2$ value and comparing models using a one-way ANOVA.

**Definitions of selective endpoints**. The lowest selective endpoint determined by statistical analysis of qPCR data was defined as the LOEC and the highest concentration where no significant selection occurred as the no observable effect concentration (NOEC). The selective endpoint determined by the selection coefficient is defined as the MSC.

**Reporting summary**. Further information on research design is available in the Nature Research Reporting Summary linked to this article.

## Data availability
The datasets associated with Figs. 1–6 are included in this published article as a Supplementary Data file. Metagenome sequence files have been deposited in the European Nucleotide Archive. Accession number: PRJEB38942.

## Code availability
Code used for metagenome analysis: FastQC; MultiQC; FLASH2; Metaphlan2; Hclust2 and ARGs-OAP v2.

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

## Acknowledgements

I.C.S. was supported by a BBSRC/AZ CASE Studentship BB/N504026/1. A.K.M. was supported by a BBSRC/AZ CASE Studentship, BB/L502509/1, and a NERC Industrial Innovation Fellowship, NE/R01373X/1. L.Z. was supported by Natural Environment Research Council grants NE/M011259/1 and NE/N019717/1. The authors would like to thank Richard Henshaw and JJ Valletta for their help and advice on statistical analysis.

## Author contributions

I.C.S., A.K.M., L.Z., J.S. and W.H.G. contributed equally to the design of the work. I.C.S. undertook the experimental work for the macrolides and tetracycline. A.K.M. undertook the experimental work for ciprofloxacin. A.K.M. undertook the bioinformatics analysis of the metagenome samples. I.C.S. drafted the paper and A.K.M., L.Z., J.S. and W.H.G. contributed to revisions and have read the final version.

## Competing interests

J.S. is an employee and shareholder of AstraZeneca PLC. All remaining authors declare no competing interests.
