## [Peer Review File · Communications Biology]

Reviewers' comments:

Reviewer #1 (Remarks to the Author):

This work represents an important assessment of the concentrations of antibiotics that are able to impact bacterial communities. As the threats posed by the spread of antibiotic resistant bacteria continue to increase, it is imperative to understand how the presence of antibiotics at a wide range of concentrations affect that spread. This work contains a detailed accounting of those effects and will inform decisions aimed at defining safe levels of antibiotics in the environment. For that reason, this work is likely to be impactful. I have specific comments and minor concerns listed below.

1. The sample collection could use more information. For instance, line 90 states the samples were frozen, but does not state a temperature.
2. Do the data in table 1 come from a literature review (implied in line 100)? If so, the sources of the data need to be cited.
3. Line 108 contains a concentration that is reported with too many significant figures. This should be adjusted to the appropriate accuracy.
4. There are instances in the methods section where the rationale for the experiments is reported (e.g. lines 100-109 and lines 115-121). These explanations should appear in the results section. Information in the methods section should be restricted to include only the information needed to replicate the experiments.
5. Line 248 reports communities dominated by E. coli. I would like to see some mention of whether this is expected or not and the potential impact of this result is.

Reviewer #2 (Remarks to the Author):

The research manuscript "Evolution of antibiotic resistance at low antibiotic concentrations: selection below the minimal selective concentration" determines the minimal selective concentrations for some antibiotics, showing that the selective potential varies among different antibiotics, even the ones from the same class.

The work has a good experimental design but needs some improvements:

L75: correct "as high as..."

L108: In line 59 authors refer that the Watch list includes macrolides, ciprofloxacin and amoxicillin. By the introduction I got the idea that the rationale was to use the antibiotics referred in the Watch list. However, tetracycline was used instead of amoxicillin. Why?

L130: qPCR instead of QPCR.

L143: Why just the samples grown with azithromycin were plated?

L153: Please explain the rationale for the selection of the samples for metagenome sequencing. Why just the samples from macrolides experiments?

L155: Please use a more formal language: "prepared" instead of "prepped"

L163: Why do the statistical analysis at a confidence level of 90%? This is too low. I suggest that 95 and 99% confidence are used. Considering the calculations of the prevalence, it was done some correction considering the concentration of the gene observed in the absence of antibiotic?

L189: The antibiotics potency was considered for the preparation of the antibiotic's solutions, right?

L239: Give a space in "it is"

L250: correct "concentration of antibiotic"

L362-363 and 388: Please revise these sentences.

Supplementary materials:

L163: The heatmap does not has error bars. Please correct the legend and revise if the same error is not in other legends.

L303: Please explain that replicas are represented.

Reviewer #3 (Remarks to the Author):

Minor comments:

1- The introduction and conclusions sections are largely and can be shortened.

2- Data of MICs of antibiotics are missing. What are the MICs of each antibiotic at day 7? Are similar to the MICs at days 0? The MICs data should be provided.

A document with track changes has been provided. Line numbers refer to the document when viewed with track changes.

Response to reviewers' comments:

Reviewer #1 (Remarks to the Author):

This work represents an important assessment of the concentrations of antibiotics that are able to impact bacterial communities. As the threats posed by the spread of antibiotic resistant bacteria continue to increase, it is imperative to understand how the presence of antibiotics at a wide range of concentrations affect that spread. This work contains a detailed accounting of those effects and will inform decisions aimed at defining safe levels of antibiotics in the environment. For that reason, this work is likely to be impactful. I have specific comments and minor concerns listed below.

	Reviewer comment	Author response
1.	The sample collection could use more information. For instance, line 90 states the samples were frozen, but does not state a temperature.	This section has been updated to include the temperature, the date of sampling, the location of the plant and that it was a grab sample.
2.	Do the data in table 1 come from a literature review (implied in line 100)? If so, the sources of the data need to be cited.	The full table from the literature review has been added to an excel spreadsheet called Supplementary Data as well as the full reference list for this. A sentence has been added to the main text to indicate this (Line 195): "Table S1 showing the full list of concentrations and the references can be found in the Supplementary data."
3.	Line 108 contains a concentration that is reported with too many significant figures. This should be adjusted to the appropriate accuracy.	This is now reported to two significant figures.
4.	There are instances in the methods section where the rationale for the experiments is reported (e.g. lines 100-109 and lines 115-121). These explanations should appear in the results section. Information in the methods section should be restricted to include only the information needed to replicate the experiments.	These sections have been moved to the appropriate results section. The section that was on lines 100-109 has been moved to appropriate sections or deleted where this was already stated in the results section. This section can now be found split across lines 192, 208, 276 and 295. What was on line 115 has been moved and adapted to fit in the results section "Assessing the selective potential of macrolides. Real-time PCR analysis" line 185). The section that reads "qnrS was the class specific gene targeted by qPCR to investigate selection by ciprofloxacin at a

		range of concentrations. It is the most common gene identified from clinical Enterobacteriaceae isolates, is mobile and is often found in environmental strains ^{21,44} . In addition, qnr genes have been reported embedded in complex class 1 integrons” is now on line 279
5.	Line 248 reports communities dominated by E. coli . I would like to see some mention of whether this is expected or not and the potential impact of this result is.	A sentence has been added in line 344 and reads: “Whilst experimental population is dominated by E. coli and Escherichia spp, there is still a diverse population of bacterial species present. This is not unexpected as the inoculum used was raw wastewater and E. coli is a faecal coliform bacterium ⁵⁰ . The laboratory conditions that these experiments were undertaken in also favour the growth conditions of E. coli and other Escherichia spp. The species are Gram negative opportunistic pathogens and are, therefore, of great concern in regards to the emergence of resistance. In addition, E. coli has been shown to be a reservoir for the macrolide resistance gene mphA ³⁵ , which was consistently the resistance gene found to be one of the most abundant genes in all three metagenome datasets.”

Reviewer #2 (Remarks to the Author):

The research manuscript “Evolution of antibiotic resistance at low antibiotic concentrations: selection below the minimal selective concentration” determines the minimal selective concentrations for some antibiotics, showing that the selective potential varies among different antibiotics, even the ones from the same class.

The work has a good experimental design but needs some improvements:

	Reviewer comment	Author response
1.	L75: correct “as high as...”	This has been corrected
2.	L108: In line 59 authors refer that the Watch list includes macrolides, ciprofloxacin and amoxicillin. By the introduction I got the idea that the rationale was to use the antibiotics referred in the Watch list. However, tetracycline was used instead of amoxicillin. Why?	The macrolides were the only antibiotics on the first watch list (2015) but ciprofloxacin was listed as a compound of concern. This work was planned and undertaken before the 2018 watchlist was published with amoxicillin on it. As mentioned in line 83 the tetracycline selection experiment was undertaken to compare the method used in this study to the method presented in Lundström et al. 2016 where they use a biofilm experiment to investigate the minimal selective concentration of tetracycline. Work on the

		MSC of ciprofloxacin has also been previously published so this also acts as a comparator.
3.	L130: qPCR instead of QPCR.	This has been changed.
4.	L143: Why just the samples grown with azithromycin were plated?	Only one of the macrolides was plated to keep the experiment manageable. Previous work in the group has shown that plating this type of culture after leaving at 40C overnight significantly changes the community structure. This plating work, therefore, had to be undertaken on the same day. As azithromycin is the most potent macrolide this was the one that was chosen to be plated as it was expected that this antibiotic was expected to induce a response at the lowest concentration of antibiotic. As previous work to determine the minimal selective concentration of ciprofloxacin and tetracycline (Kraupner et al. 2018 and Lundström et al. 2016) has been undertaken previously, the main focus of this work was on the macrolide antibiotics with the other two antibiotics as comparisons to previous studies. A sentence has been added (line 136): "As azithromycin has been shown to be more potent than the natural compound, erythromycin²⁸ and has a lower MIC²⁹, a plating experiment was only conducted on azithromycin as it is expected that it will show the lowest response of the three macrolides."
5.	L153: Please explain the rationale for the selection of the samples for metagenome sequencing. Why just the samples from macrolides experiments?	As explained in response to the previous comment, there has been no work previously investigating the macrolide antibiotics, as with tetracycline and ciprofloxacin and therefore more extensive work was undertaken with these antibiotics to determine the lowest concentration at which selection is observed which included the metagenomics sequencing. Also, as we did not see selection for resistance until relatively high macrolide concentrations we thought it was important to investigate the potential selection of other macrolide resistance genes at lower concentrations using metagenomics.

		A sentence has been added on line 242. “Metagenome analysis was undertaken on a subset of samples from macrolide selection experiments as this was the main focus of the study and hasn’t been investigated by previous studies. Metagenome analysis of tetracycline and ciprofloxacin selection has, however, been previously investigated in studies by Lundström et al. 2016 and Kraupner et al. 2018. Three replicates were taken from each treatment including the LOEC for all three macrolides (750 µg/L), a concentration below this (250 µg/L) and concentrations higher than this where a strong selective effect is seen by intI1 (1,000, 10,000 and 100,000 µg/L).”
6.	L155: Please use a more formal language: “prepared” instead of “prepped”	This has been corrected.
7.	L163: Why do the statistical analysis at a confidence level of 90%? This is too low. I suggest that 95 and 99% confidence are used. Considering the calculations of the prevalence, it was done some correction considering the concentration of the gene observed in the absence of antibiotic?	The 90% confidence level was chosen because of the mixed community experiment. Mixed community experiments are inherently noisy and it was, therefore, decided that 90% confidence would highlight less strong associations between antibiotic concentration and resistance gene prevalence where biological effects may occur. From an environmental perspective, this generally allows for a more protective and conservative LOEC (and subsequently NOEC and PNEC) value to be determined. This will, therefore, be more protective at preventing selection in the environment. A sentence clarifying this has been added into line 161 and reads: “As mixed community experiments are inherently noisy due to founder effects and stochasticity, 90% confidence was also highlighted to show less strong associations and will be more protective of selection for AMR in the environment. ” The calculation of prevalence was undertaken to adjust for the vastly different number of bacteria at day 0 and day 7. At day 0 the bacterial count is approximately 10³ per ul (wastewater influent dilute in broth) and at day 7 (after growth overnight) the bacterial count is approximately 10⁸ per ul. This difference in density of bacteria will greatly affect the abundance of resistance

		genes and it is, therefore, meaningless to compare the abundance of resistance genes at day 0 and day 7. Prevalence corrects for bacterial cell count. This does not correct for the concentration of gene between the presence or absence of antibiotic, the statistical analysis investigates if there is a statistical significance between the two.
8.	L189: The antibiotics potency was considered for the preparation of the antibiotic's solutions, right?	For ciprofloxacin a vast range of concentrations were tested spanning many orders of magnitude and the potency of ciprofloxacin was, therefore, not taken into account when determining the concentration range. For tetracycline, as this was a comparison to a previously published study, potency was not taken into account when deciding on concentration range as this was based on the concentration range used in previously published work. Potency was taken into account when deciding on the concentration range to test for the range finding experiments of the macrolide antibiotics. As erythromycin is the natural compound and there is evidence to suggest that its semi synthetic derivatives are more potent, the range finding experiments for erythromycin went an order of magnitude higher than those for azithromycin and clarithromycin. This is stated in line 185.
9.	L239: Give a space in "it is"	This has been corrected
10.	L250: correct "concentration of antibiotic"	This has been corrected
11.	L362-363 and 388: Please revise these sentences.	These two sentences now read: "The MIPC is important as concentrations of antibiotic above this will decrease the rate at which resistance bacteria disappear from the environment. This will result in an increase human exposure risk and the probability of subsequent evolution in comparison to environments where no antibiotics are present. It is less of a concern, however, than if positive selection was occurring where numbers of resistance genes and resistant bacteria increase over time." "However, with increased persistence the number of resistant bacteria or resistance genes will decrease over time at antibiotic concentrations above the MIPC whereas

		enrichment through positive selection (at antibiotic concentrations above the MSC) will lead to increased AMR over time, so the two phenomena are fundamentally different in terms of outcome.”
12.	Supplementary materials: L163: The heatmap does not has error bars. Please correct the legend and revise if the same error is not in other legends.	This sentence has been removed from the three heatmaps and added into Figure S28 where it was previously missing.
13.	Supplementary materials: L303: Please explain that replicas are represented.	This has been amended for all three diversity plots.

Reviewer #3 (Remarks to the Author):

Minor comments:

	Reviewer comment	Author response
1.	The introduction and conclusions sections are largely and can be shortened.	Both of these have been shortened as much as possible without removing any critical content. This can be seen via track changes in the manuscript.
2.	Data of MICs of antibiotics are missing. What are the MICs of each antibiotic at day 7? Are similar to the MICs at days 0? The MICs data should be provided.	Data of MICs was not established. Traditionally MIC values are calculated using single species of bacteria. In this study we have exclusively used mixed communities in the experiments. MIC values were, therefore, not determined for these studies.

Updated figures:

Bar graphs from Figures 1, 3, 4 and 6 have been reformatted into boxplots. For panel graphs, A, B etc. have been changed to lowercase letters. Commas have been added in for numbers >1,000.

Figure 3:

Figure 4:

Figure 6:

a**b**
REVIEWERS' COMMENTS:

Reviewer #1 (Remarks to the Author):

I have read through this manuscript, and I am satisfied that my concerns have been adequately addressed.

Reviewer #2 (Remarks to the Author):

All my concerns were clearly clarified and all the suggestions and corrections correctly addressed. I recommend the publication.

I have just to minor corrections to ask:

L87. Include the units "-80 °C"

L301 and 304: Correct "Enterobacteriaceae"

Reviewer #3 (Remarks to the Author):

The authors have responded to my comments.